# Research on Multi-Scale Ecological Network Connectivity—Taking the Guangdong–Hong Kong–Macao Greater Bay Area as a Case Study

**DOI:** 10.3390/ijerph192215268

**Published:** 2022-11-18

**Authors:** Jiansheng Wu, Shengyong Zhang, Haihao Wen, Xuening Fan

**Affiliations:** 1Key Laboratory for Urban Habitat Environmental Science and Technology, School of Urban Planning and Design, Peking University, Shenzhen 518055, China; 2Key Laboratory for Earth Surface Processes, Ministry of Education, College of Urban and Environmental Sciences, Peking University, Beijing 100871, China

**Keywords:** ecological networks, ecological connectivity, complex networks, ecological groups, Guangdong–Hong Kong–Macao Greater Bay Area

## Abstract

The Guangdong–Hong Kong–Macao Greater Bay Area urban agglomeration is an urban agglomeration with some of the most intensive urbanization since 1980s. A large amount of cultivated land, forest land, water bodies and other land types in the region has been occupied by construction land, resulting in fragmented ecological landscapes and biodiversity in the region and causing many other ecological problems. Based on this, this paper takes the Guangdong–Hong Kong–Macao Greater Bay Area as a case study, constructs an ecological network of the dispersion scale of five species from 1990 to 2020 based on a morphological spatial pattern analysis (MSPA) method, identifies the ecological groups in the network and uses the core node-based community evolution path tracking algorithm to analyze the ecological groups in order to explore the changes of ecological network connectivity at different scales in the region and to reveal the overall and local characteristics and changes of the migratory space of terrestrial mammals with different dispersion capabilities. The research results show that: (1) From 1990 to 2020, the area of construction land in the Guangdong–Hong Kong–Macao Greater Bay Area increased sharply, with good connectivity in the northwest, southwest and eastern regions and poor connectivity in the central region. (2) There are obvious differences between the overall and local changes in the connectivity trends of multi-scale regional ecological networks. On the whole, the overall ecological connectivity of the ecological network at each scale showed a gradual upward trend, and the overall connectivity index IIC and the possible connectivity index *PC* gradually increased with the increase of the maximum dispersal distance of species. From the perspective of local patches, the larger the species dispersion scale, the larger the value of the revised betweenness centrality index and the patch possible connectivity index. (3) The distribution of ecological groups at different species dispersion scales is different, and the smaller the dispersal scale of the species, the greater the distribution of ecological groups. Small-scale species are limited by the maximum dispersal distance, and the range of their ecological groups is generally small. Small-scale (3 km), mesoscale (10 km) and large-scale (30 km) core nodes of ecological groups show a gradual increase trend, and the overall connectivity of ecological groups has improved. However, the core nodes of the extra-large-scale (60 km) and ultra-large-scale (100 km) ecological groups show a trend of decreasing fluctuations, and the overall connectivity within the ecological group has declined. This study is helpful to clarify the structural characteristics of regional ecological space and provide a theoretical basis for regional ecological planning.

## 1. Introduction

Since the reform and opening up, China’s urbanization process has been rapid and the social economy has developed rapidly, but at the same time, its ecological security situation has become increasingly severe. The scale of urbanization in a region or country affects the process of its modernization, and urban space continues to occupy more natural ecological space, causing the decline of regional ecosystem connectivity, fragmentation of internal landscapes, and damage to ecological functions, directly exacerbating the contradiction between man and nature [1]. Massive regional population migration, high-intensity land development and the disorderly expansion of urban construction often lead to the continuous disappearance and fragmentation of natural landscapes [2]. Landscape fragmentation affects the exchange and flow of material and energy between ecosystems and the migration and diffusion of organisms and changes in the structure of green spaces [3,4,5], resulting in a series of weakened ecosystem services, loss of biodiversity and frequent occurrence of natural disasters. Environmental problems hinder the sustainable development of the region. Maintaining the functional safety and biodiversity of regional natural ecosystems has become a topic of great concern to scholars around the world [6].

In response to the landscape fragmentation problem caused by rapid urbanization, ecological networks have gradually become the focus of urban ecological security research and spatial planning [7,8,9,10]. Ecological network is a comprehensive spatial configuration method, which comprehensively considers biodiversity conservation, habitat maintenance and other issues [11], and its core goal is to achieve the efficient flow of energy, material and information in the ecosystem [12]. This method aims to identify key ecological patches and ecological corridors by comparing the importance of different landscape patches to ecological processes and ecological functions in the entire region, and to form a relatively stable and functionally effective biological habitat network system to achieve maintenance, or restoring regional landscape connectivity [13,14]. Maintaining or restoring landscape connectivity is one of the most effective solutions for biodiversity conservation in fragmented landscape environments [15,16]. Good landscape connectivity can facilitate the migration and dispersal of organisms, gene exchange and other key ecological processes [17]. Conversely, a decline in landscape connectivity may lead to the isolation of biological populations, a decline in regional species abundance and an increase in inbreeding, thereby increasing the risk of species extinction [18].

Maintaining the long-term stability of biodiversity is one of the important goals of ecological civilization construction, but there are still certain challenges in protecting a variety of biological species on large temporal and spatial scales, and different species have different needs for the spatial configuration of habitats [19]. Because species have different minimum habitat area requirements and different mobility capabilities, ecological networks applicable to one species or group of species may not well reflect the mobility status of other species [20]. In addition, research on complex networks has found that many networks of certain practical research significance can be naturally divided into multiple communities or modules, that is, densely connected vertex groups in the network but only sparse connections between groups [21]. In ecological networks, they also have corresponding structural characteristics, which are called ecological groups [22]. According to the theory of island biogeography, large habitat patches can accommodate more species than small habitat patches and thus can serve as “species pools” for surrounding small habitat patches [23,24]. “Species pools” generate species associations in surrounding small habitat patches, creating radiation effects [25,26]. Because there are usually multiple large patches in the ecological environment as “species pools” affected by radiation effects, fragmented landscapes form multiple ecological groups in space. These ecological groups have the same properties as the concept of “community” in complex networks, being tightly connected within themselves but loosely connected to each other [27,28]. In addition, the state of an ecological group also changes with changes in its internal habitat patches, and these changes lead to a change in the state of the ecological group, which in turn affects the entire ecological network. The evolution of ecological groups reveals the local characteristics of ecological networks to a certain extent [29] and provides a theoretical basis for regional ecological planning.

A large amount of cultivated land, forest land and water bodies in areas that have been occupied by construction land, resulting in the fragmentation of ecological landscape and the reduction of biodiversity in the area [30]. In response to the negative impact of rapid urban agglomeration development on ecosystems, ecological networks have become the focus of research on urban ecological security and territorial spatial planning [7,8,31]. Therefore, this study takes the Guangdong–Hong Kong–Macao Greater Bay Area as an example to construct an ecological network of the dispersion scale of five species from 1990 to 2020 based on the morphological spatial pattern analysis (MSPA) method. The community evolution path tracking algorithm based on core nodes is used to analyze the evolution path of ecological groups, so as to explore the changes of ecological network connectivity at different scales in the region, clarify the dispersion status of species at different scales in the ecosystem and reveal the local characteristics of ecological networks. The technology roadmap is shown in Figure 1. This study aims to answer the following two core questions: (1) What are the differences in ecological network connectivity and its spatiotemporal changes at different species dispersion scales? (2) Considering the local differences in ecological network connectivity, should the spatial distribution and changes of ecological groups be studied at different species dispersion scales? The specific research contents of this study are as follows: (1) The analysis of land use change and landscape fragmentation degree. (2) The temporal and spatial changes of ecological network connectivity at different species dispersion scales. (3) The spatial distribution and changes of ecological groups at different species dispersion scales.

## 2. Materials and Methods

### 2.1. Study Area and Data Sources

The Guangdong–Hong Kong–Macao Greater Bay Area is located between 21°–25° northern latitude and 111°–116° eastern longitude and encompasses the Hong Kong Special Administrative Region, Macao Special Administrative Region and nine cities in the Pearl River Delta (Guangzhou, Shenzhen, Dongguan, Foshan, Zhuhai, Huizhou, Zhongshan, Jiangmen, Zhaoqing), with a total area of 55,080 km^2^, as shown in Figure 2. In 2019, the Central Committee of the Communist Party of China and the State Council issued the “Guangdong–Hong Kong–Macao Greater Bay Area Development Plan”, which clearly stated that in the future, the Guangdong–Hong Kong–Macao Greater Bay Area should not only become a dynamic world-class city cluster, an international technology innovation center and in-depth cooperative between the mainland and Hong Kong and Macau, but that the area should also be built into a high-quality living region that is livable, business-friendly and travel-friendly and becomes a model of high-quality development.

The basic data and sources used in this study can be seen in Table 1. The land use data comes from the first Chinese annual land cover data from Landsat produced by Professor Yang Jie and Huang Xin of Wuhan University on the Google Earth Engine (GEE) [32]. Set (CLDC) with a spatial resolution of 30 m and a time span from 1990 to 2020. The source of DEM data is Geospatial Data Cloud (http://www.gscloud.cn/, (accessed on 1 May 2022)). Species distribution data come from the IUCN Red List of Threatened Species (https://www.iucnredlist.org/, (accessed on 1 May 2022)).

### 2.2. Methods

#### 2.2.1. Species Habitat Identification

##### Regional Species Dispersion Scale Determination

Using the known distribution data of the IUCN Red List of Threatened Species, this study extracted a total of 90 species of terrestrial mammals existing in the Guangdong–Hong Kong–Macao Greater Bay Area. According to the research data of Harestad and Bunnell [33], Hoffman [34], Grassman [35] and Rajaratnam [36], 13 terrestrial mammal categories with clear habitat range were further identified. In addition, it was found in previous studies that the maximum dispersal distance of mammalian species is proportional to the size of the habitat range [37]. Cécile Tannier and Marc Bourgeois et al. concentrate on forest mammals living in the woodland patches and using the farmland areas for hunting or movement. Ten forest mammal species found in their study area and threatened by urbanization were identified using a regional naturalist data base and the IUCN Red List. These species were subdivided into three groups (small, medium and large mammals) according to two main functional traits, the home range and the dispersal distance. From the three groups, three emblematic protected species were selected, the Eurasian lynx (Lynx lynx), the wild cat (Felis silvestris) and the red squirrel (Sciurus vulgaris), and the connectivity differences of their network of nature reserves were analyzed [38]. Distance from these two main functional characteristics subdivided 13 terrestrial mammal classes into 5 groups representing 5 different scales of species dispersal scale: (1) small scale, suitable for small habitat range (10 hectares) and terrestrial mammals with a short maximum dispersal distance (3 km); (2) mesoscale, suitable for terrestrial mammals with a small habitat range (60 hectares) and a species with a short maximum dispersal distance (10 km); (3) large scale, suitable for terrestrial mammals with medium habitat range (300 hectares) and medium maximum dispersal distance of species (30 km); (4) extra-large scale, suitable for large habitat range (500 hectares) and the largest species terrestrial mammals with wide dispersal distances (60 km); and (5) ultra-large scale, suitable for terrestrial mammals with a large habitat range (1000 hectares) and species with a wide maximum dispersal distance (100 km). Instead of using a single species to represent a specific group, the study used whole numbers to represent their general relationship (Appendix A Appendix A).

##### Fragmentation Analysis Based on Morphological Spatial Pattern Analysis (MSPA) Method

This study used MSPA to identify ecological sources [39,40]. MSPA is based on a series of mathematical morphological principles, including erosion, dilation, opening operation, closing operation, etc., to analyze the geometry and connectivity of land cover patterns in raster images [41] from 1990, 2000 and 2010, and based on the land use data of the Guangdong–Hong Kong–Macao Greater Bay Area in 2020, forest land, shrubs and grasslands were extracted as the foreground of MSPA, and other landscape types were used as the background to conduct MSPA analysis on the landscape structure of the Guangdong–Hong Kong–Macao Greater Bay Area in four periods.

#### 2.2.2. Construction of Potential Ecological Corridors

##### Construction of Resistance Surface Based on Landscape Resistance

Considering factors, such as natural conditions and human disturbance, and referring to the relevant literature [33], based on the actual situation of the Guangdong–Hong Kong–Macao Greater Bay Area and the availability of data, this study selected elevation, slope, land use type and distance. A comprehensive resistance index system was then constructed for the four indicators of the distance of construction land, and the classification and resistance value range of each resistance factor were determined through the expert consultation method and reference to the relevant literature [42,43]. The resistance values of the four resistance factors was set between 1 and 1000, where the larger the resistance value, the greater the obstacles encountered by the species in the process of migration and dispersion. The weight and assignment of the resistance factor is shown in Table 2.

After that, using ArcGIS 10.4.1 software(Environmental Systems Research Institute, Redlands, CA, USA), the multi-factor comprehensive evaluation method was used to superimpose and analyze four types of resistance surfaces, such as land use, distance from construction land, elevation and slope. Combined with the grid calculator, the comprehensive resistance surface is calculated, as shown in Figure 3, and the calculation formula is:(1)$$\;R_{Final}=R_{LULC}\times 0.32+R_{DTC}\times 0.27+R_{S}\times 0.22+R_{E}\times 0.19\;$$
where $R_{Final}$ represents the final comprehensive resistance value, $R_{LULC}$ represents the resistance value of the land use type resistance factor, $R_{DTC}$ represents the resistance value of the distance resistance factor from the construction land, $R_{S}\;$ represents the resistance value of the slope resistance factor and $R_{E}$ represents the resistance value of the elevation resistance factor value.

##### Potential Ecological Corridor Analysis

The minimum cumulative resistance model (minimum cumulative resistance, *MCR*) was first proposed by Kaaapen et al. Essentially, it reflects the trend of species migration and the level of connectivity between patches [12,44]. It can be used to construct ecological corridors to increase material and energy flow between source sites [45]. The specific solution formula is as follows:
(2)$$MCR=f_{min}\sum_{j=n}^{i=m}D_{ij}\times R_{i}$$
where $f_{min}$ represents the minimum cumulative resistance value, $D_{ij}$ represents the distance between species from ecological source patch i and ecological source patch  j and $R_{i}$ represents the corresponding ecological source patch i presents to a species movement resistance.

The *MCR* model was used to quantitatively identify the ecological network corridors in the Guangdong–Hong Kong–Macao Greater Bay Area. Graphab 2.6 software (CNRS—Université Bourgogne Franche-Comté, Besançon, France) was used to input the corresponding minimum area and maximum dispersion distance thresholds based on the species dispersion scale. The comprehensive resistance surface and ecological source constructed above were used. According to the identification results of the land (MSPA core area), the minimum path between each ecological source area is generated. According to the actual situation inside the study area, considering that ecological corridors are often combined with planning in practice, the extraction of corridors should avoid crossing the ecological source as much as possible. This study chose to ignore the corridors across the source areas, that is, no corridors between the other two source areas intersecting with the intermediate source areas and finally constructed the potential ecological network of the Guangdong–Hong Kong–Macao Greater Bay Area.

#### 2.2.3. Ecological Network Connectivity Evaluation

##### Overall Connectivity Analysis of Ecological Network

The degree of network closure (*α* index), line point rate (*β* index) and network connectivity (*γ* index) can be used to measure the integrity and complexity of the network structure (as in Equation (3)). The greater the degree of network closure, the more loops appear and the more paths that can be dispersed. The line point rate represents the average number of connections in the ecological source. *Β* < 1 indicates that the network is a tree structure, *β* = 1 indicates that the network is a single loop structure and *β* > 1 indicates that the network connection is more complex. The higher the degree of network connectivity, the higher the connectivity of ecological sources in the ecological network [46].
(3)$$\alpha =\frac{L-V+1}{2V-5};\beta =\frac{L}{V};\gamma =\frac{L}{3(V-2)}$$
where L is the number of ecological corridors in the ecological network and V is the number of ecological sources.

##### Ecological Source Connectivity Analysis

In this study, two indicators were used to evaluate the connectivity of each ecological source patch: the modified betweenness centrality index $BC^{PC}$ and the patch possible connectivity index dPC to analyze the effect of a single ecological source patch on the overall ecological network Contribution of connectivity.

The modified betweenness centrality index $BC^{PC}$ index evaluates the connectivity of each ecological source. This index is integrated by the betweenness centrality index BC and the patch importance index “dPC connector”, which takes into account the patch area and species dispersion probability and use the maximum product probability to define the shortest path between nodes [47]. From the perspective of landscape ecology, the modified betweenness centrality index $BC^{PC}$ includes the area product and the shortest path weighting, which is conceptually compatible with the characteristics of the connectivity index *PC*. The calculation formula of the modified betweenness centrality index $BC^{PC}$ is as follows [48]:
(4)$$\begin{array}{c} {BC}_{i}^{PC}=\sum_{j}\sum_{k}a_{j}a_{k}e^{-\alpha d_{\mathrm{jk}}}\\ j,k\in \{1..n\},k<j,i\in P_{\mathrm{jk}} \end{array}$$
where $BC^{PC}$ represents the sum of all the shortest paths through patch i and each path is weighted by the product of the area of the connected patch and its species dispersion probability and $P_{jk}$ represents the difference between patch j and patch k. The shortest path in-between passes through all patches. The parameter α is determined by two other parameters: distance d and probability p, calculated as follows:(5)α=−log(p)/d 
where p is the probability of a species moving between two ecological sources, d is the distance between these ecological sources and the parameter α represents the decreasing speed of the moving probability p as the distance d increases. The average dispersion distance of species corresponds to the distance d between ecological sources when p = 0.5, while the maximum dispersion distance of species corresponds to the distance d between ecological sources when p = 0.05 or 0.01. In the study, the minimum cumulative cost distance between ecological sources was used as the distance d between ecological sources to represent the mobility of actual species within the region and p was set to 0.05.

The patch possible connectivity index “dPC“, which measures the connectivity loss at the patch scale of each ecological source area, that is, quantifies the impact of each ecological source patch on the overall connectivity after the removal of each ecological source patch. The calculation formula is as follows:(6)$$dPC_{i}=100\times \frac{PC-PC_{\mathrm{removei}\;}}{PC}\;$$
where $dPC_{i}$ represents the possible connectivity index of patch i, PC is the overall possible connectivity index of the ecological network before removing the ecological source patch i and $PC_{\mathrm{removei}}$ is the ecological network after removing the ecological source patch i. Regarding the overall possible connectivity index, a high value of dPC indicates high importance.

##### Ecological Corridor Connectivity Analysis

The research uses indicators based on graph theory and circuit theory to evaluate the connectivity of ecological corridors in ecological networks, which can effectively analyze the length, quality and importance of ecological corridors to quantify all the ecological corridors in the network. Interactions between patch pairs and the role of each corridor in mediating ecological flows. The quality of ecological corridors is quantified by corridor width and the ratio of minimum cost path distance to path length (CWD_LCPL). The CWD_LCPL indicator reflects the relative resistance value on each path, representing the movement of species along the minimum cost path between patch pairs average resistance encountered. The larger the value, the worse the quality of the ecological corridor. The corridor width represents the physical width of the resistance diffusion corridor, that is, the ratio of the corridor area to the corridor length. The current flow metric (CF) based on circuit theory [49], also called current betweenness centrality by Carroll [50], refers to the frequency with which species cross corridors in an ecological network [51], which is calculated as follows:(7)$$CF_{i}=\sum_{j}^{n}c_{i}^{j}$$
where $CF_{i}$ represents the sum of currents passing through patch i, $c_{i}^{j}$ is the current passing through patch i when the current is sent from all patches (except j) to patch j and patch j is connected to the ground.

#### 2.2.4. Multi-Scale Ecological Group Evolution Research

##### Multiscale Ecological Group Identification

For ecological networks that connect species habitats, communities in the network correspond to ecological groups that are closely related to habitats in the area. Identifying this community structure in the ecological network is helpful to understand the characteristics of the biological groups in the entire region and to explore the interconnections of regional ecological groups. This study uses Graphab 2.6 software to identify communities in the Guangdong–Hong Kong–Macao Greater Bay Area ecological network. The optimal modularity method adopts the most common greedy clustering algorithm. The greedy clustering algorithm starts with a singleton cluster, iteratively merges the two clusters and produces the cluster with the best modularity, that is, chooses the largest increase or the smallest decrease and proceeds to n-1 clusters after merging, where the result returns the cluster that achieves the highest modularity [21]. While modularity is a measure of the quality of graph node clustering, the basic principle is that good clustering involves a large number of links within a group and a small number of links between groups. The community structure can be precisely searched by finding network partitions with positive, numerically large modularity values.

Suppose G=(V,E) is an undirected connected graph with n:=|V| vertices and m:=|E| edges. Let C={C1,…,Ck} denote the partition of V. C is the cluster of G and Ci of the non-empty set. The subgraph of G represents identifying a cluster Ci, namely G[Ci]:=(Ci,E(Ci)), where E(Ci):={{v,w}∈E:v,w∈Ci}. Then, E(C):=∪i=1kE(Ci) is the set of intra-cluster edges and E\E(C) is the set of inter-cluster edges. The number of intra-cluster edges is denoted by m(C), and the number of inter-cluster edges is denoted by m¯(C). The set of edges with one endpoint in Ci and the other endpoint in Ci is denoted E(Ci,Cj). Then the modularity index q(C) of cluster C is defined as [21]:
(8)q(C):=∑C∈C[|E(C)|m−(|E(C)|+∑C∈C|E(C,C′)|2m)2]


In the research, the calculation formula of the edge weight $E_{ij}$ is as follows:(9)$$E_{ij}=a_{i}a_{j}e^{-\alpha d_{ij}}$$
where the parameter α represents the importance of the ecological source area $a_{i}a_{j}$ to the edge weight.

##### Ecological Node Importance Calculation and Ecological Group Evolutionary Path Tracking Algorithm

This study evaluates the importance of ecological nodes in the ecological network and uses this as the weight value of the corresponding nodes in the ecological network to identify the core nodes of the ecological group. Referring to the study of Luo [22], the importance of patch location and the importance of habitat function are considered to form a patch importance index. The positional importance of each ecological source was evaluated by modifying the betweenness centrality index $BC^{PC}$, and the size of each ecological source patch was calculated as the evaluation index of habitat function importance. After the metrics were normalized to the [0, 1] range, patch location importance and habitat function importance were synthesized according to the following formula:(10)PNI=0.5∗CPI+0.5∗HFI
where *PNI* is the importance of ecological nodes, *CPI* is the importance of patch location and *HFI* is the importance of habitat function. *PNI* is the weight value *W* of the corresponding node in the following Formula (11), that is, according to the formula 10, the core nodes of the group in the ecological network can be further identified as the basis for the evolution analysis of the ecological group.

This study adopts the algorithm of tracking the evolution path of community core nodes to analyze the evolution path of ecological groups in the ecological network of Guangdong–Hong Kong–Macao Greater Bay Area. The algorithm steps are as follows:
(1)Distinguish core nodes and non-core nodes in the ecological group.

The way to judge whether a node is a core node is to compare its weight difference with all neighboring nodes, the formula is as follows:
(11)$$Cen(v)=\sum_{v_{i}\in A_{v}}W_{v}-W_{v_{i}}$$
where Cen(v) is the centrality index of node v. If its value is greater than 0, the node is a core node, otherwise it is a non-core node. $W_{v}$ is the weight of node v, and $W_{v_{i}}$ is the weight of the neighbor nodes of node v. This method does not need to set parameters.

(2)For ecological groups $C_{t}$ and $C_{t+1}$ at time t and time t+1, if the core node of $C_{t}$ appears in $C_{t+1}$, then $C_{t}$ and $C_{t+1}$ have an evolution relationship.

##### Ecological Group Life Cycle Model

Observing and analyzing the behavior of ecological groups in the evolution paths of different ecological groups is helpful to analyze the impact of species in the actual ecological environment by emergencies or the development trend of ecological groups in the region. Referring to Zhang Weiwei’s [52] research on community evolution, he defined seven types of ecological group formation, ecological group disappearance, ecological group expansion, ecological group shrinkage, ecological group stability, ecological group split and ecological group merger. Evolutionary behavior and the use of ecological group predecessors to determine ecological group behavior.

## 3. Results

### 3.1. Multi-Scale Ecological Network Construction

#### 3.1.1. Important Habitat Distribution

After MSPA analysis, the ecological landscape of the Guangdong–Hong Kong–Macao Greater Bay Area is divided into seven types, and the results are shown in Figure 4.

The landscape types of the study area based on MSPA had a large quantitative change in 2000, while other time periods were relatively stable, as shown in Figure 4 and Figure 5. During the period from 1990 to 2020, the area of the core area of the Guangdong–Hong Kong–Macao Greater Bay Area fluctuated, and the proportion of the core area in the total prospective area decreased from 87.54% in 1990 to 87.10% in 2000 and remained stable in 2010 and 2020 at around 87.10%. In terms of distribution, the core area is located in the northwest, southwest and east of the Guangdong–Hong Kong–Macao Greater Bay Area, with good spatial connectivity. In the central area, there are also a few core area patches, but the distribution is relatively scattered and has poor connectivity, which hinders the material exchange of biological species to a certain extent; the proportion of the area shows a diametrically opposite trend to that of the core area. After rising from 3.01% in 1990 to 5.61% in 2020, it gradually increased to 6.11% in 2010 and 2020. Pores also accounted for 3.01% of the total prospect area in 1990 and stabilized at around 2.55% from 2000 to 2020. The area of the edge region and pores is second only to the area of the core region. The branch line accounted for 1.95% of the total area in 1990 and continued to decrease to 0.83% in 2020, indicating that within the study area, the branch line played a certain degree of connection; the bridge area in 1990 and the total prospective area. The proportion was 1.46% and continued to decrease to 0.59% in 2020. The area of the bridging area is not large and continues to decline. In the case of species migration, it is difficult to realize the flow and exchange of energy and materials as an ecological corridor. Isolated islands exist in isolation in the entire ecological network and can act as stepping-stones. They are fragmented in the study area, accounting for 1.50% of the total area in 1990 and will drop to 0.44% in 2020. The area is relatively small. The ring road area provides a path for the group migration of animals in the entire ecological network and facilitates the internal migration of species.

Based on the above research on the regional species dispersion scale in the Guangdong–Hong Kong–Macao Greater Bay Area, the minimum habitat area corresponding to the five species dispersion scales was studied and used as the screening threshold for the core area extracted by MSPA analysis. Each scale ecological network includes all sources that meet the minimum habitat requirements for mammals at that scale (10 ha for small-scale mammals, 60 ha for mesoscale mammals, 300 ha for large-scale mammals, 500 ha for extra-large-scale mammals and 1000 ha for ultra-large-scale mammals).

#### 3.1.2. Spatial Distribution of Ecological Corridors

From 1990 to 2020, the potential corridors in the Guangdong–Hong Kong–Macao Greater Bay Area were closely connected with each other and the ecological network was good, but there were certain differences in the ecological sources, potential corridors and average corridor lengths of the five scattered scales, as shown in Appendix A in the Appendix A. From 1990 to 2020, the overall spatial distribution of potential corridors in the Guangdong–Hong Kong–Macao Greater Bay Area was consistent, as shown in Figure 6 and Figure 7, and Appendix A. On a small scale, potential corridors are mainly concentrated in the west and east and are scattered in the middle. Since the maximum dispersion distance of species is short, there are obviously more ecological corridors in the northwest and eastern regions with good ecological effects. While in the central area with dense built-up areas, although there is a lack of good ecological effects, there are still some corridors that can connect urban green spaces. Potential corridors at the mesoscale are mainly concentrated in the east and southwest and are scattered in the middle. Since the maximum dispersion distance of species is short, there are not many ecological corridors in the northwest and northeast regions with good ecological effects due to the existence of large continuous ecological sources. While the ecological effects in the southeast and southwest are relatively good, the source areas are relatively fragmented. There are obviously more ecological corridors in the area. In addition, although there is a lack of good ecological effects in the dense central area of built-up areas, there are still a small number of corridors that can connect urban green spaces. On a large scale, potential corridors are mainly concentrated in the southwest and scattered in the east and south. Since the maximum dispersion distance of species is medium, the ecological corridors are significantly reduced in the northwest and northeast regions with good ecological effects due to the existence of large continuous ecological sources, while the ecological corridors in the southwest have better ecological effects and more broken source areas. In addition, there are very few corridors in the dense central area of the built-up area. At the extra-large scale, potential corridors are mainly concentrated in the southwest and scattered in the east. Since the maximum dispersion distance of species is large, the northwest and northeast regions with good ecological effects have few ecological corridors due to the existence of large continuous ecological sources. While the ecological effects in the southwest are better, the ecological corridors are significantly more fragmented in the source areas and there are almost no corridors in the dense central area of built-up areas. At the ultra-large scale, potential corridors are mainly concentrated in the southwest and scattered in the southeast. Since the maximum dispersion distance of species is large, the northwest and northeast regions with good ecological effects have few ecological corridors due to the existence of large continuous ecological sources. While the ecological effects in the southwest are better and the ecological corridors are more obvious in the areas with more fragmented sources, there are no corridors in the dense central area of the built-up area.

### 3.2. Ecological Network Connectivity Evaluation

#### 3.2.1. Overall Connectivity Analysis of Ecological Network

During the period from 1990 to 2020, there were certain differences in the change trends of network closure, line point rate and network connectivity, as shown in Table 3. From the perspective of network closure, the small-scale and mesoscale ecological networks have little annual change and the average value is close to zero or even negative, indicating that in the ecological network with a small species dispersion scale, there are fewer loops and the ecological sources can be diffused. Paths are extremely limited. In large-scale, extra-large-scale and ultra-large-scale ecological networks, the network closure decreased significantly during 1990–2000 and then slowly recovered during 2000–2020. The larger the species dispersion scale, the higher the corresponding network closure, that is, with the scale of dispersal becomes larger, the closed loops in the ecological network gradually increase. The closed loops can open up a shortcut for the transfer of corresponding species and increase the diffusible paths of ecological sources.

From the point of view of the line point rate, in the small-scale and mesoscale ecological networks, the line point rate is less than 1 most of the time, and only the line point rate of the mesoscale ecological network is greater than 1 in 2020, indicating that the ecological networks of these two scales are basically. The above is a tree-like structure, and the network structure is relatively simple. In the large-scale, extra-large-scale and ultra-large-scale ecological networks, the line point rate decreased significantly in 2000 and remained basically unchanged from 2000 to 2020, and the larger the species dispersion scale, the higher the corresponding line point rate. That is to say, as the scale of species dispersion becomes larger, the connections between sources in the ecological network increase and the network structure becomes more and more complex.

From the perspective of network connectivity, the results are different from the network closure and line point rate. The ecological network of each species dispersion scale shows a gradual upward trend from 1990 to 2020, and the larger the species dispersion scale, the larger the corresponding network. The lower the degree of connectivity, that is, as the scale of species dispersion becomes larger, the connectivity of ecological sources in the ecological network becomes worse.

#### 3.2.2. Ecological Source Connectivity Analysis

The modified betweenness centrality index $BC^{PC}$ and the patch possible connectivity index “dPC” basically showed the law that the larger the species dispersion scale, the larger the value, as shown in Table 4. The revised betweenness centrality index $BC^{PC}$ in small-scale, mesoscale, large-scale and extra-large-scale ecological networks gradually decreased during 1990–2010 and increased again in 2020, while in the ultra-large-scale ecological networks, the value shows a continuous downward trend. The larger the scale of species dispersion, the higher the value of the modified betweenness centrality index $BC^{PC}$, but there is an exception, that is, in 2010, the modified betweenness centrality index $BC^{PC}$ of the large-scale ecological network showed a poor highest performance and its value was lower than that of the mesoscale ecological network. The modified betweenness centrality index $BC^{PC}$ represents the role of the ecological source patch as a stepping-stone in the entire ecological network, indicating that in 2010, the ecological source patch played a stepping-stone role in the ecological network. Generally low, it resulted in a decline in the local connectivity of the ecological network.

The value of the patch possible connectivity index “dPC” fluctuated and decreased during the 1990–2020 period in small-scale, mesoscale and large-scale ecological networks, while it remained relatively stable in ultra-large-scale and ultra-large-scale ecological networks. It declined during the 2000–2020 period but showed a sustained and slow upward trend in the period of 2000–2020. An ecological network with a larger species dispersion scale has a larger patch possible connectivity index “dPC” value, that is, an ecological network with a larger species dispersion scale generally has a greater degree of importance to the overall network connectivity of its ecological source patches.

#### 3.2.3. Ecological Corridor Connectivity Analysis

The changes in the connectivity indicators of ecological corridors at various scales in the Guangdong–Hong Kong–Macao Greater Bay Area from 1990 to 2020 are shown in Table 5. During 1990–2020, the CWD_LCPL values of the ecological network at different species dispersion scales showed a uniform trend and gradually increased during the period of 1990–2010, that is, the relative resistance value of the corridors in the ecological network gradually increased and the quality of the corridors decreased year by year, which reflects that the rapid urbanization process of the Guangdong–Hong Kong–Macao Greater Bay Area caused a certain level of environmental pressure on the biological migration space in the area, but the CWD_LCPL value tends to be stable during the period of 2010–2020, which also reflects the ecological protection work in this area. In addition, from different scales, the value of CWD_LCPL in the small-scale ecological network is the lowest and the relative resistance of the corridor in the ecological network is the smallest; while the value of CWD_LCPL in the large-scale, extra-large-scale and ultra-large-scale ecological network is close and at a high level, and the ecological the relative resistance of the corridors in the network is larger. The average width of corridors in small-scale and mesoscale ecological networks continued to decrease during 1990–2020, while the values in large-scale, extra-large-scale and ultra-large-scale ecological networks showed a decreasing trend during 1990–2010. During the period of 2010–2020, it rebounded again, but the rebound was not large, which reflected that the ecological corridors in the ecological network of various scales were being infested to varying degrees, and the biological migration space was forced to shrink. Of these, the small-scale and mesoscale ecological networks are the most prominent. The mean CF varies greatly in scale. During the period from 1990 to 2020, the mesoscale and large-scale ecological network showed a U-shaped trend, while the small-scale, extra-large-scale and ultra-large-scale ecological network showed a fluctuating and declining trend. The scale of the ecological network has the highest mean CF, which shows that the probability of organisms migrating between habitats is the largest at this scale.

### 3.3. Multiscale Ecological Group Evolutional Group Analysis Results

#### 3.3.1. Results of Ecological Group Analysis

In 2020, the distribution of ecological groups at the five dispersion scales in the Guangdong–Hong Kong–Macao Greater Bay Area is different, and the smaller the species dispersion scale, the more ecological groups are distributed, as shown in Figure 8. Species with a small dispersal scale are limited by the maximum dispersal distance, and the range of their ecological groups is generally small, especially in urban agglomerations. Ecological groups in urban areas are mostly composed of isolated single or several small ecological sources, which are numerous and scattered; ecological groups in non-urban areas are larger in scope and consist of multiple large ecological sources and their surrounding scattered small ecological sources. The composition of ecological sources.

Among the five dispersion scales, Jiangmen, Huizhou, and Zhuhai have large-scale ecological sources in the Guangdong–Hong Kong–Macao Greater Bay Area, but there is much cultivated and construction land at the junction of urban and rural areas in Jiangmen and Huizhou. The ecological sources in the region are separated. Due to the accumulation of human activities along the coast of Zhuhai, the ecological sources in the region are separated, resulting in many fragmented ecological groups. Species in the urban area are greatly affected by human activities. The distribution is small and few, the actual species dispersion distance is small, and they can only move between a single or several nearby fragmented ecological sources; Zhaoqing and Huizhou have large contiguous ecological sources, and the ecological sources in the region are well connected. There are few divided ecological sources in the region, forming ecological groups with ecological sources clustered together. Shenzhen, on the other hand, has a smaller area than other cities in the Guangdong–Hong Kong–Macao Greater Bay Area. In addition, the large-scale ecological sources in the area are distributed evenly. The ecological sources are well connected, and species exchanges between different sources are relatively smooth.

#### 3.3.2. Ecological Group Analysis Results

The statistics of the number of ecological groups and core nodes extracted from the ecological network of each scale in the Guangdong–Hong Kong–Macao Greater Bay Area from 1990 to 2020 are shown in Table 6. The study found that the smaller the study scale, the greater the number of ecological groups in the ecological network, which is affected by the maximum dispersal distance of species. Compared with large-scale species, small-scale species have a small range of movement and can only move between different habitats that are close to each other. In the ecological network, the ecological corridors between the close ecological sources have a large weight and are closely connected, while the ecological corridors between the long-distance ecological sources have a small weight, and the connections are sparse or even impossible to connect together, resulting in the ecological network in the ecological network. The distribution of groups is more dispersed and the number increases.

For small-scale and mesoscale species, the ecological groups in the ecological network showed a gradual increasing trend from 1990 to 2020, indicating that the living spaces of small-scale and mesoscale species have gradually dispersed in the past three decades. The overall internal connectivity has improved. For large-scale species, this trend is not obvious and the living space distribution of the species is relatively stable, while the core nodes of the ecological group show a trend of increasing fluctuations, and the overall internal connectivity of the ecological group is not obviously significantly improved. For extra-large-scale species, the ecological groups in the ecological network showed a small increasing trend from 1990 to 2020, the living space of their species was still developing in a more dispersed direction, and the core nodes of the ecological group showed a decrease in fluctuations. The overall connectivity of ecological groups has declined; however, the living space of super-large species shows greater stability. From 1990 to 2020, the number of ecological groups in the ecological network has remained basically unchanged. It shows that large-scale species have greater adaptability to changes in the living environment, while the core nodes of ecological groups show a gradual decrease trend and the overall connectivity within ecological groups decreases significantly.

#### 3.3.3. Ecological Group Core Node Analysis Results

In the ecological network, the number of ecological group stabilization events at different species dispersion scales is relatively large, as shown in Figure 9. This shows that no matter what scale of species dispersion, most of the ecological groups in the ecological network are in a relatively stable state, and there are many ecological groups that can maintain multiple time periods, and the regional ecosystem has a certain stability. However, there are still some differences in the number of evolutionary events of other ecological groups at different species dispersal scales. On a small scale, the overall number of ecological groups is relatively large, and the evolution events of ecological groups focus on the formation and disappearance of ecological groups. The reasons for the frequent occurrence of ecological group evolution events may be related to the area of ecological groups and the rapid urbanization in the region. Because the area of small-scale ecological groups is generally small, the core nodes in the ecological group are easily affected by human activities and change, which makes most ecological groups unable to effectively find their precursor or successor ecological groups, resulting in the continuous formation and development of ecological groups. At the mesoscale ecological group due to the increase of its area, the phenomenon slowed down significantly. At the large-scale and above ecological group, the number of ecological group evolution events occurred significantly less, which may be due to the fact that at this scale. The ecological group has a large area, and the ecological nodes in the ecological group are more and more stable, which leads to a low level of the number of ecological group evolution events. This phenomenon is most obvious in the ultra-large-scale ecological group. The number of ecological group evolution events is zero.

Taking the small-scale ecological network as an example, the evolutionary events of the formation and disappearance of ecological groups mainly occurred at the urban–rural junction of Jiangmen City and Huizhou City. Human activities have a great influence, causing the core nodes of ecological groups to change frequently, and ecological groups cannot find effective precursor or successor ecological groups, resulting in the continuous formation of new ecological groups and the disappearance of old ecological groups. The Hong Kong–Shenzhen Dapeng ecological group is composed of core nodes in Shenzhen and Hong Kong, and large ecological sources are connected together in patches, with good connectivity between ecological sources. During the period from 1990 to 2020, the group experienced evolutionary events, such as ecological group expansion, stability and division. During the period from 1990 to 2010, the development trend of the Hong Kong–Shenzhen Dapeng ecological group was good, and the number of core nodes in the group increased, the range and area tended to be stable, but during the period from 2010 to 2020, the connection between the internal nodes of the ecological group and the peripheral ecological space nodes weakened, and they gradually separated into independent ecological groups, resulting in the split of ecological groups.

Compared with the small-scale ecological network, the range and area of ecological groups in other-scale ecological networks are larger, and the evolution events of their ecological groups are also different in different periods, as shown in Figure 10.

## 4. Discussion

### 4.1. Connectivity Risk Assessment

This study focuses on the local differences in ecological connectivity between species habitats in the ecological network and identifies ecological groups in ecological networks at different scales based on this. To explore changes in the local connectivity of ecological networks at different scales in the region. The network model regards the stepping-stones as the nodes of the network and the possible migration paths of species as the edges of the network [53]. Connecting important habitat patches is an important part of effectively maintaining biodiversity. Risk management is the ultimate goal of risk assessment. In the future, it will be particularly important to manage and control risks based on the identification of stepping-stone risk structures. Considering the risk causal chain model, we should focus on the dominant role of exposure and vulnerability on connected risks, classify stepping-stones with medium and high risks in the medium-level security landscape, discuss the stepping-stone management under different security patterns and analyze the stepping-stones that need to be protected under different patterns.

### 4.2. Analysis of Ecological Group Differences at Multiple Scales

Ecological planning should be carried out on the multiple scales of biological dispersal at the same time and evaluating the spatial distribution of multi-scale ecological networks is helpful to comprehensively clarify the dispersal status of different species in the ecosystem in the region. Considering the overall and local connectivity differences of the ecological network, core nodes are extracted, ecological groups are identified and the evolution analysis of ecological groups is carried out. In this study, thirteen terrestrial mammal classes were subdivided into five groups, representing five different scales of the species dispersal scale: small scale, mesoscale, large scale, extra-large scale and ultra-large scale. There are certain differences among the five scattered scale ecological networks. At the same scale, even if the number of ecological sources are not significantly different, the causes may be different. For example, at small scales, the ecological group data in Zhongshan and Foshan are different and the causes are the same, but at the mesoscale, the number of groups in Zhongshan and Zhaoqing are the same, while the causes are different. Due to the wide range of human activities, Zhongshan is concentrated and distributed in patches, so that the ecological groups in its urban area are only composed of a single space or very few. It consists of a small ecological source area, but the ecological group has a large area and a small number and is distributed in the urban area. However, the actual reflection is that the species in the urban area are greatly affected by human activities, and the ecological source area is less distributed. The actual species dispersal distance is small, and they can only move between a single or several nearby fragmented ecological sources; Zhaoqing has a large ecological source that is connected as a block, the ecological sources in the region are well connected and the area is divided. There are few open ecological sources, and ecological groups have been formed, in which ecological sources are clustered together.

### 4.3. Analysis of Ecological Group Evolution Events at Multiple Scales

At the urban–rural junction of Jiangmen City and Huizhou City, the ecological source area within the region is generally small. There is much cultivated land and construction land at the urban–rural junction of Jiangmen and Huizhou. The scattered construction land separates the ecological sources in the region, causing frequent changes in the core nodes of the ecological group. The ecological group cannot find an effective precursor or successor ecological group, resulting in the continuous formation of new ecological groups and the disappearance of old ecological groups. The Hong Kong–Shenzhen Dapeng Ecological Group is composed of core nodes in Shenzhen and Hong Kong, and large ecological sources are connected together in a single piece, with good connectivity between ecological sources. The group experienced evolution events, such as the expansion, stability and division of ecological groups from 1990 to 2020. The development trend of the Hong Kong–Shenzhen Dapeng ecological group was good from 1990 to 2010. The core nodes in the group increased, and the scope and area became stable. However, the connection between the internal nodes of the ecological group and the external ecological space nodes weakened during 2010 to 2020, and the ecological group gradually separated into independent ecological groups, causing the division of the ecological group. Hong Kong and Shenzhen are greatly affected by the intensity of human activities, and the ecological groups are also greatly affected. The change of ecological groups in Jiangmen, Huizhou, Shenzhen and Hong Kong will not only affect the stability of species themselves but also interfere with the implementation of local policies, which is not conducive to the stable implementation of policies. Policymakers should pay attention to the evolution of ecological groups when planning.

### 4.4. Potential Applications of the Planning Section

The development of regional urbanization has led to drastic changes in the surface environment, causing serious damage to regional biodiversity and various changes in the distribution of species that inhabit it. By analyzing the evolution characteristics of ecological groups and their spatial distribution, this study helps to understand the characteristics of biological groups in the entire region. The 13 species of terrestrial mammals selected in this study can be divided into two categories, scale-sensitive and scale-insensitive. It can be found that organisms with fixed range of activities, such as Lepus sinensis, Moschus berezovskii Flerov and Martes flavigula, belong to the scale-sensitive type, similar to Sus scrofa, Paradoxurus hermaphroditus Pallas, Prionailurus bengalensis and other creatures with a wide range of activities and that are scale-insensitive. Clarify the spatial structure of ecological land and provide theoretical basis for regional ecological planning.

### 4.5. Research Limitations and Future Prospects

There are still some deficiencies in this study, which need to be further improved: the extraction of ecological sources needs to be further refined. The spatial morphological MSPA method chosen in this study is too general for the selection of suitable habitats for terrestrial mammals and does not consider the differences in suitable habitats among different species. In the future, the multi-scale suitable habitat characteristics in the region should be integrated to clarify the actual suitable habitat distribution of species; the setting of the resistance surface should improve the objectivity. The scores and weights of different indicators in the resistance surface refer to the results of previous studies, but there is still a certain degree of subjectivity. In the future, by further confirming the habit characteristics of multiple species in the region, the InVEST model can be used to comprehensively consider the multi-factor status of the regional ecology and the influencing factors of ecological connectivity changes need to be clarified. In the future, we can focus on a variety of human activity indicators to explore their impact on changes in regional ecological network connectivity, so as to explore how to effectively maintain or enhance the landscape connectivity of protected species under the pressure of human activities.

## 5. Conclusions

This study takes the Guangdong–Hong Kong–Macao Greater Bay Area, where the urbanization process is most intensive in China, as an example, in order to reveal the local characteristics and changes of the migratory space of terrestrial mammals at different species dispersion scales. The main conclusions are as follows: The area of construction land in the Guangdong–Hong Kong–Macao Greater Bay Area increased sharply, while the area of forest land fluctuated and decreased slowly. The area of forest land in the suburbs remained basically unchanged, and the area of forest land in the urban area of the Guangdong–Hong Kong–Macao Greater Bay Area changed significantly. The MSPA method shows that the core area of the Guangdong–Hong Kong–Macao Greater Bay fluctuates, mainly in the northwest, southwest and east and has good connectivity, while the central area has only a few core areas, which are relatively scattered with poor connectivity. There are obvious differences between the overall and local changes of regional ecological network connectivity trends at different species dispersion scales. On the whole, from 1990 to 2020, the overall ecological connectivity of the ecological network at each scale showed a gradual upward trend, but there were still subtle differences: from the local individual point of view, the ecological network with a larger scale of species dispersion, the correction of the intermediate centrality index $BC^{PC}$ value and the “dPC” value of the possible connectivity index of plaques were larger, but in 2010, the large-scale ecological network revised betweenness centrality index $BC^{PC}$ performed the worst. The overall number of ecological groups with smaller species dispersion scale was larger, and the evolution events of ecological groups were concentrated on the formation and disappearance of ecological groups. In contrast, the number of ecological group evolution events with a larger species dispersion scale was smaller. In the future, small-scale species will still be greatly affected by urbanization, so we need to pay more attention to the protection of small-scale species.

## Figures and Tables

**Figure 1 ijerph-19-15268-f001:**
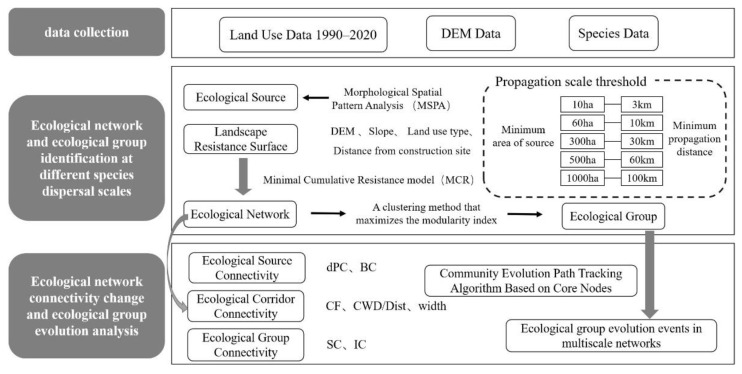
Technology Roadmap.

**Figure 2 ijerph-19-15268-f002:**
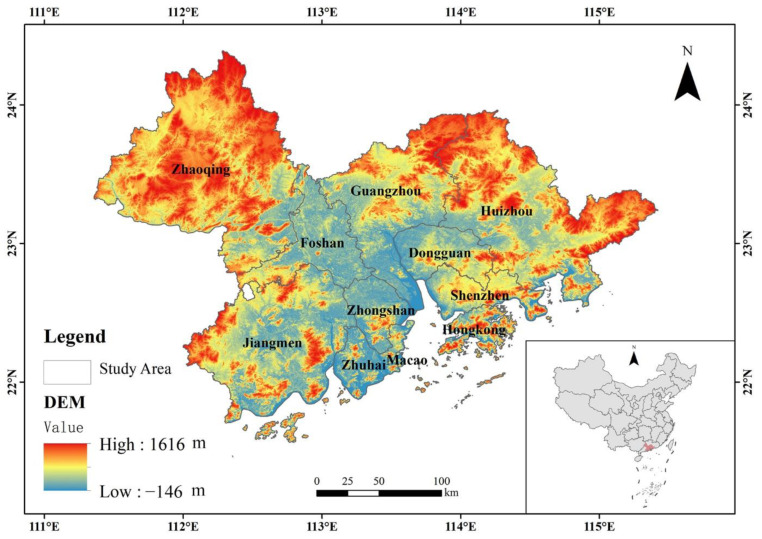
Geographical location of study area.

**Figure 3 ijerph-19-15268-f003:**
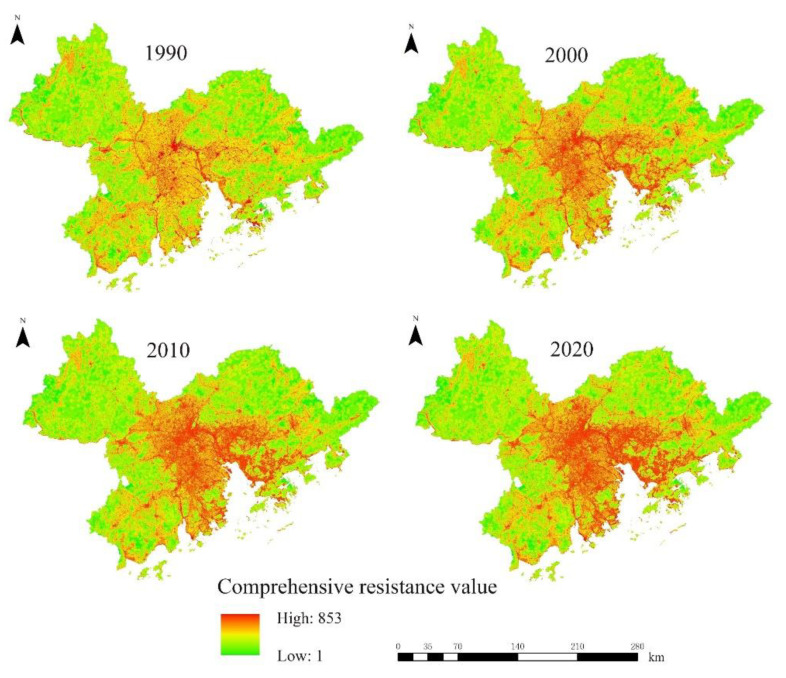
The final comprehensive resistance value from 1990 to 2020.

**Figure 4 ijerph-19-15268-f004:**
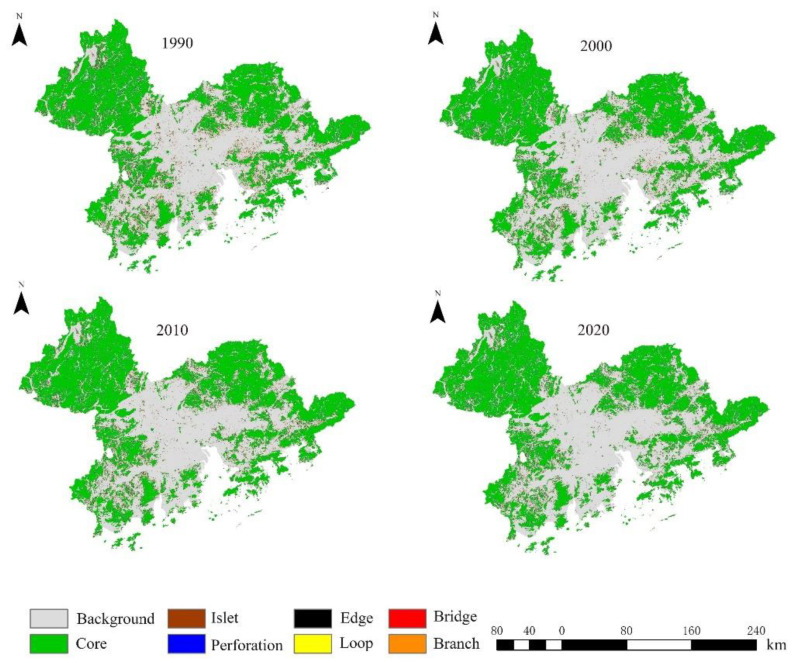
Landscape type map of the study area based on MSPA.

**Figure 5 ijerph-19-15268-f005:**
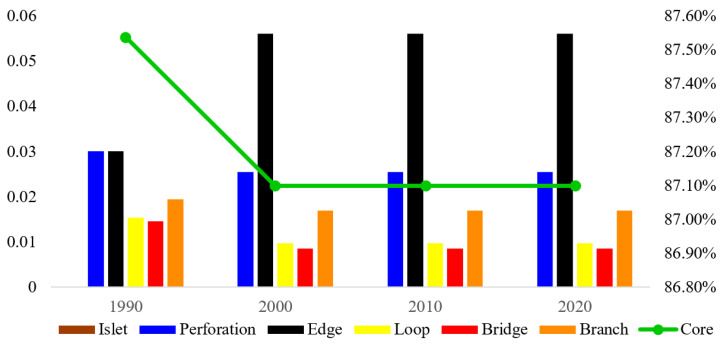
Changes in the proportion of MSPA landscapes in the total prospective area from 1990 to 2020.

**Figure 6 ijerph-19-15268-f006:**
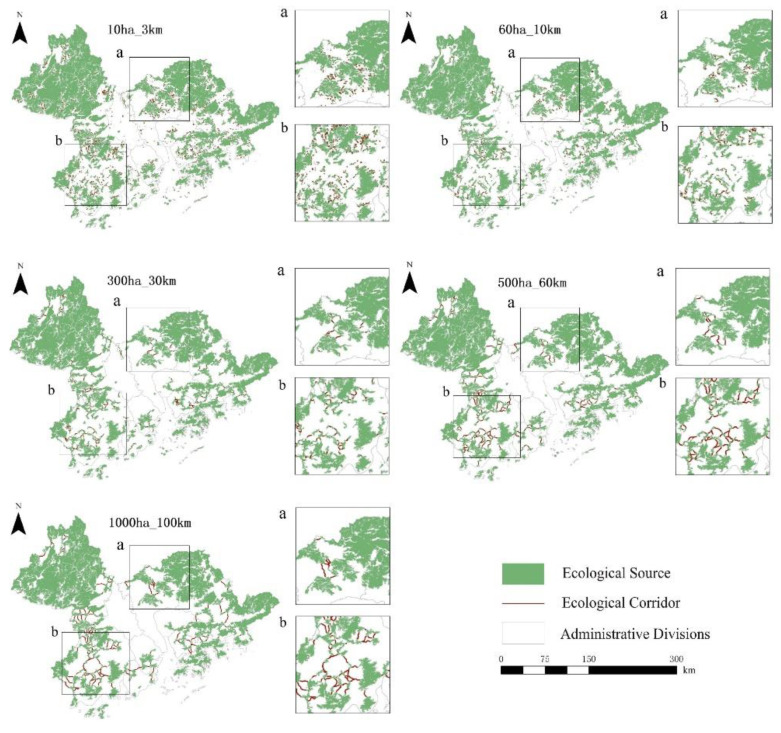
Five Scattered Scale Ecological Networks in the Guangdong–Hong Kong–Macao Greater Bay Area in 1990.

**Figure 7 ijerph-19-15268-f007:**
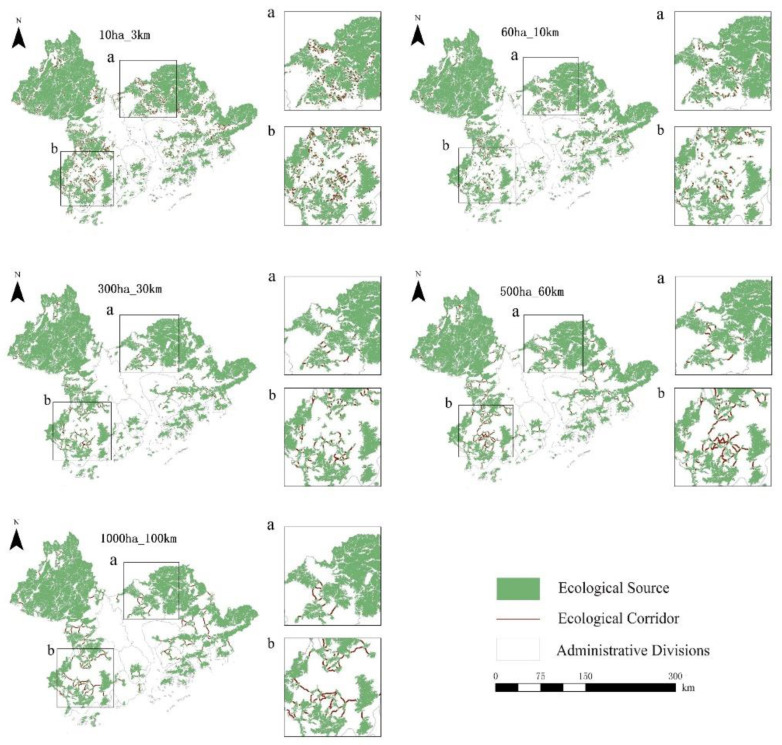
Five Scattered Scale Ecological Networks in the Guangdong–Hong Kong–Macao Greater Bay Area in 2020.

**Figure 8 ijerph-19-15268-f008:**
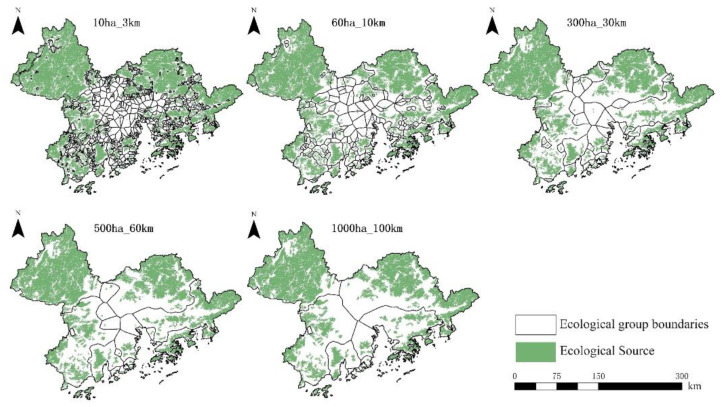
Five scattered scale ecological groups in the Guangdong–Hong Kong–Macao Greater Bay Area in 2020.

**Figure 9 ijerph-19-15268-f009:**
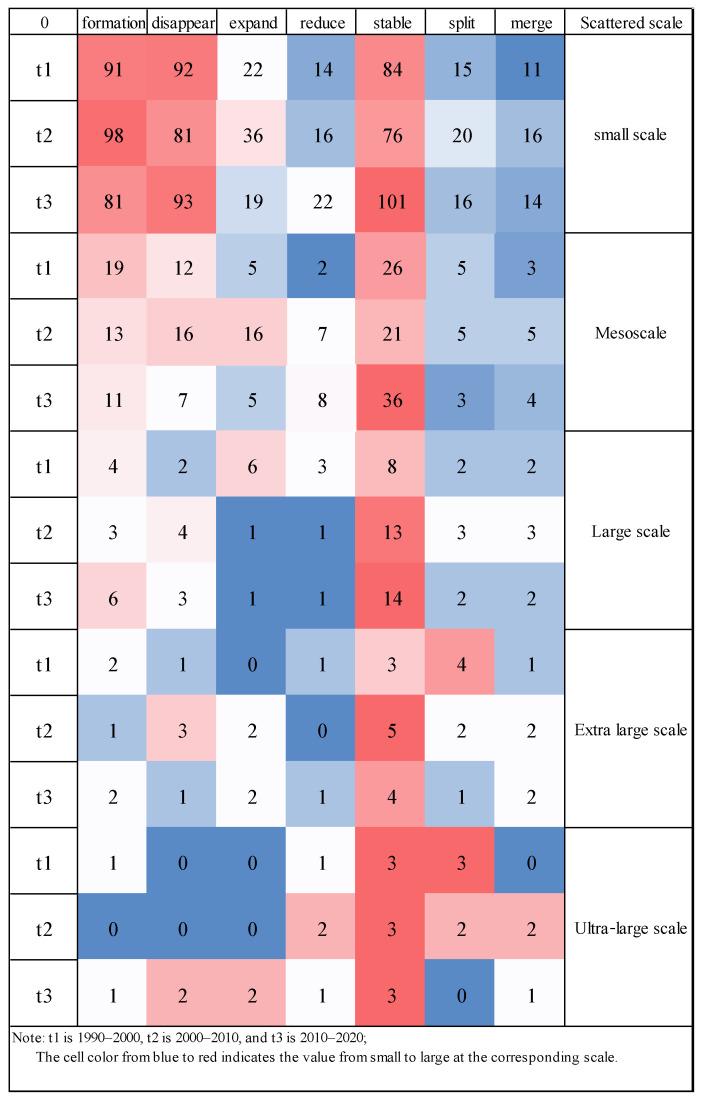
Number of evolutionary events of various ecological groups.

**Figure 10 ijerph-19-15268-f010:**
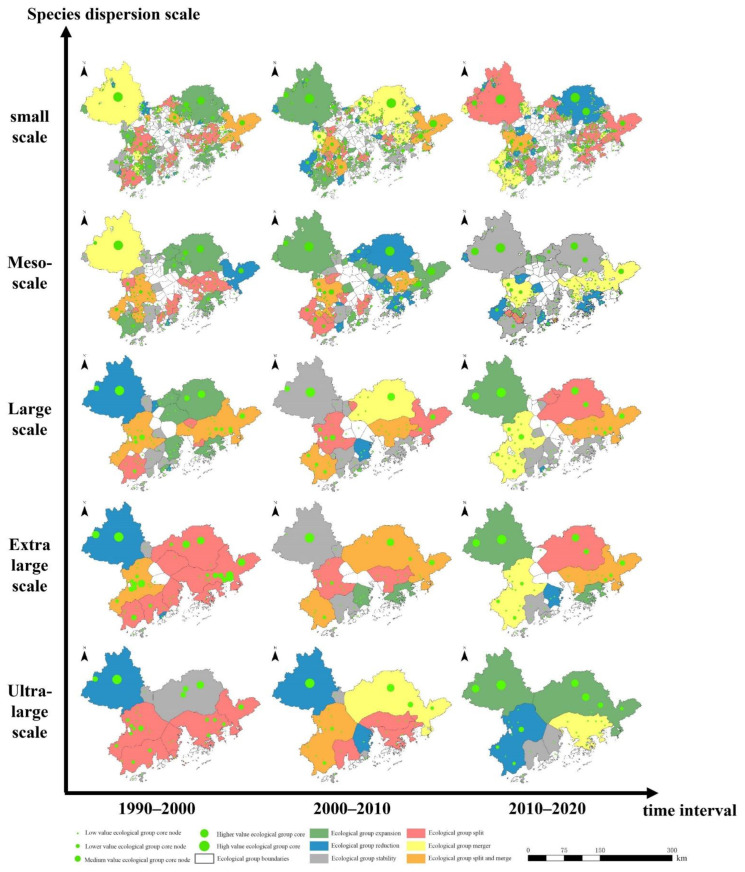
Distribution of evolutionary events of various ecological groups.

**Table 1 ijerph-19-15268-t001:** Basic data list.

Data Name	Format and Resolution	Year	Data Source
land use data	30 m tiff	1990–2020	http://doi.org/10.5281/zenodo.4417809, (accessed on 1 May 2022)
DEM data	30 m tiff	2000–2009	http://www.gscloud.cn/, (accessed on 1 May 2022)
species distribution data	shp	2008–2021	https://www.iucnredlist.org/, (accessed on 1 May 2022)

**Table 2 ijerph-19-15268-t002:** Resistance factor assignment and weight.

Resistance Factor	Category	Weights	Resistance Assignment
land use type	construction land	0.32	1000
	water		900
	unused land		600
	farmland		500
	grassland		100
	shrub		1
	forest		1
Slope (°)	0–15	0.22	1
	15–30		200
	30–45		400
	>45		800
Elevation (m)	≤100	0.19	1
	100–200		200
	200–400		400
	>400		600
Distance to construction land (m)	≤200	0.27	900
	200–500		700
	500–1000		500
	1000–2000		300
	>2000		1

**Table 3 ijerph-19-15268-t003:** Changes in α, β and γ indices from 1990 to 2020.

Species Maximum Dispersal Distance	Network Index	1990	2000	2010	2020
3 km	α	−0.084	−0.118	−0.112	−0.096
	β	0.832	0.764	0.775	0.807
	γ	1,822,293	1,754,581	2,025,543	2,408,752
10 km	α	0	0	−0.006	0.013
	β	0.999	0.999	0.987	1.025
	γ	174,484	197,377	198,656	243,346
30 km	α	0.120	0.075	0.079	0.092
	β	1.234	1.145	1.152	1.179
	γ	28,667	27,412	27,590	33,042
60 km	α	0.261	0.214	0.217	0.229
	β	1.511	1.417	1.423	1.447
	γ	17,985	18,518	16,768	18,912
100 km	α	0.358	0.266	0.270	0.290
	β	1.695	1.514	1.521	1.562
	γ	9546	9475	7059	8661

**Table 4 ijerph-19-15268-t004:** Changes in ecological source connectivity indicators from 1990 to 2020.

Species Maximum Dispersal Distance	Connectivity Index	1990	2000	2010	2020
3 km	dPC mean (median)	4.75 × 10^−4^ (2.20 × 10^−8^)	4.18 × 10^−4^ (1.01 × 10^−8^)	3.71 × 10^−4^ (9.54 × 10^−9^)	3.84 × 10^−4^ (1.18 × 10^−8^)
	BC mean (median)	5.84 × 10^14^ (0)	5.51 × 10^14^ (0)	1.72 × 10^14^ (0)	4.69 × 10^14^ (0)
10 km	dPC mean (median)	1.88 × 10^−3^ (1.48 × 10^−6^)	1.50 × 10^−3^ (6.81 × 10^−7^)	1.40 × 10^−3^ (1.03 × 10^−6^)	1.45 × 10^−3^ (7.80 × 10^−7^)
	BC mean (median)	1.15 × 10^16^ (0)	6.93 × 10^15^ (0)	4.90 × 10^15^ (0)	7.91 × 10^15^ (0)
30 km	dPC mean (median)	5.39 × 10^−3^ (2.96 × 10^−5^)	5.38 × 10^−3^ (9.81 × 10^−5^)	4.19 × 10^−3^ (1.40 × 10^−5^)	4.38 × 10^−3^ (1.78 × 10^−5^)
	BC mean (median)	1.37 × 10^17^ (0)	5.97 × 10^16^ (0)	2.98 × 10^14^ (0)	8.54 × 10^16^ (0)
60 km	dPC mean (median)	7.83 × 10^−3^ (1.09 × 10^−4^)	6.17 × 10^−3^ (5.83 × 10^−5^)	6.18 × 10^−3^ (6.14 × 10^−5^)	6.66 × 10^−3^ (5.53 × 10^−5^)
	BC mean (median)	4.19 × 10^17^ (0)	2.49 × 10^17^ (2.90 × 10^14^)	6.80 × 10^16^ (0)	2.69 × 10^17^ (0)
100 km	dPC mean (median)	1.19 × 10^−2^ (3.86 × 10^−4^)	9.76 × 10^−3^ (1.96 × 10^−4^)	1.02 × 10^−2^ (2.17 × 10^−4^)	1.05 × 10^−2^ (2.05 × 10^−4^)
	BC mean (median)	5.79 × 10^17^ (5.65 × 10^15^)	4.96 × 10^17^ (3.79 × 10^15^)	4.28 × 10^17^ (5.78 × 10^15^)	3.85 × 10^17^ (2.06 × 10^15^)

Note: The data in the table is displayed using scientific notation.

**Table 5 ijerph-19-15268-t005:** Changes in the connectivity indicators of ecological corridors from 1990 to 2020.

Species Maximum Dispersal Distance	Index	1990	2000	2010	2020
3 km	quantity	2133	2006	2171	2416
	CWD_LCPL	5.64	6.39	6.56	6.75
	Average width (m)	3535.61	2444.86	2510.35	2210.49
	CF mean	715.74	682.02	401.10	462.39
10 km	quantity	724	770	768	866
	CWD_LCPL	6.32	7.08	7.39	7.65
	Average width (m)	24,703.07	19,153.41	18,861.34	18,821.88
	CF mean	833.26	592.79	820.49	814.51
30 km	quantity	327	308	310	343
	CWD_LCPL	6.78	7.62	8.02	8.10
	Average width (m)	156,048.98	139,446.57	117,817.05	128,884.65
	CF mean	552.33	295.89	560.69	536.61
60 km	quantity	287	282	269	288
	CWD_LCPL	6.72	7.57	8.06	8.14
	Average width (m)	506,442.07	427,184.24	347,738.74	390,902.57
	CF mean	540.98	589.73	309.09	424.63
100 km	quantity	222	209	181	203
	CWD_LCPL	6.54	7.42	7.97	8.04
	Average width (m)	1,179,694.60	1,124,838.41	790,221.52	946,413.21
	CF mean	303.01	273.68	220.67	234.96

**Table 6 ijerph-19-15268-t006:** Statistics on the characteristics of ecological groups in ecological networks.

Years	Species Dispersion Scale	Number of Ecological Groups	The Number of Core Nodes of the Ecological Group
1990	Small scale (3 km)	806	424
	Mesoscale (10 km)	175	121
	Large scale (30 km)	54	55
	Extra-large scale (60 km)	23	44
	Ultra-large scale (100 km)	12	35
2000	Small scale (3 km)	930	436
	Mesoscale (10 km)	192	149
	Large scale (30 km)	54	69
	Extra-large scale (60 km)	27	49
	Ultra-large scale (100 km)	14	33
2010	Small scale (3 km)	994	442
	Mesoscale (10 km)	195	142
	Large scale (30 km)	53	59
	Extra-large scale (60 km)	28	39
	Ultra-large scale (100 km)	14	26
2020	Small scale (3 km)	1019	446
	Mesoscale (10 km)	207	148
	Large scale (30 km)	60	66
	Extra-large scale (60 km)	28	41
	Ultra-large scale (100 km)	12	30

## Data Availability

The basic data and sources used in this study can be seen in Table 1. The land use data come from the first Chinese annual land cover data from Landsat produced by Professor Yang Jie and Huang Xin of Wuhan University on Google Earth Engine (GEE). Set (CLDC) with a spatial resolution of 30 m and a time span from 1990 to 2020. The source of the DEM data is Geospatial Data Cloud (http://www.gscloud.cn/ (accessed on 1 May 2022)). Species distribution data come from the IUCN Red List of Threatened Species (https://www.iucnredlist.org/ (accessed on 1 May 2022)).

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
