# Peer review of "Research on Multi-Scale Ecological Network Connectivity—Taking the Guangdong–Hong Kong–Macao Greater Bay Area as a Case Study"

_ijerph, 2022, doi:10.3390/ijerph192215268_

Round 1
Reviewer 1 Report (Previous Reviewer 2)
With appreciation to the authors, the revised MS is now improved for publication. All comments and suggested revisions have been incorporated in the revised MS.
The revised MS can be considered for publication.
Author Response
Please see the attachment.

Reviewer 2 Report (Previous Reviewer 3)
Dear Editor
the manuscript is on a acceptable status.
Kind Regards
Author Response
Please see the attachment.

Reviewer 3 Report (Previous Reviewer 4)
Accept in present form
Author Response
Please see the attachment.

Reviewer 4 Report (New Reviewer)
I am sending a few comments which, in my opinion, should be introduced to the text before its final publication.
1. Note to subsection 1. Introduction
Another works very similar to your research have already been carried out in other parts of Far East or World. In my opinion, there should be references to the following works:
- Nor A.M.N., Corstanje R., Harris J.A., Grafius D.R., Siriwardena G.R., 2017, Ecological connectivity networks in rapidly expanding cities, Heliyon 3(6)
https://doi.org/10.1016/j.heliyon.2017.e00325
- Nor A.M.N., Corstanje R., Harris J.A., Brewer T., 2017, Impact of rapid urban expansion on green space structure, Ecological Indicators 81: 274-284
http://dx.doi.org/10.1016/j.ecolind.2017.05.031
- Fenu G., Pau P.L., 2018, Connectivity analysis of ecological landscape networks by cut node ranking, Applied Network Science () 3:22
https://doi.org/10.1007/s41109-018-0085-0
- Vimal R., Mathevet R., Thompson J.D., 2012, The changing landscape of ecological networks, Journal for Nature Conservation 20(1): 49-55.
https://doi.org/10.1016/j.jnc.2011.08.001
2. Note to Figure 2.
The legend of the figure shows height values from –146 to 1616 m. Are you sure???
Could you make the background in meters a.s.l.???
3. Note to subsection 2.2. Methods
The article distinguishes 5 groups representing 5 different scales of species dispersal scale: (1) small, (2) mesoscale, (3) large, (4) extra-large and (5) ultra-large (pages 5-6).
In my opinion, it would be good to give one example of terrestial species representing each group, who live in this area. This would give more clarity to the potential reader, when he will reads the subsection 4.4. Potential Applications of the Planning Section. In this subsection two separate groups (scale-sensitive and scale-insensitive) have been separated with examples of species.
4. Note to Figure 7.
The red color for Ecological Corridors MUST BE stronger. It is now hard to see.
5. Note to Conclusion.
In point (3) you wrote:
“Evolutionary differences of multi-scale ecological groups. The distribution of ecological groups at the dispersion scale of the five species in the Guangdong-Hong Kong-Macao Greater Bay Area is different. Species with a small dispersal scale are limited by the maximum dispersal distance, and the range of their ecological groups is generally small, especially in urban agglomerations”.
Can you add one sentence about the impact of the trend of urbanization changes and the development of the current situation of these groups in the future?
Best regards
Author Response
Please see the attachment.

Reviewer 5 Report (New Reviewer)
It is an innovative research with novel ideas, rich content, sufficient data and fluent text, but there are still several questions to be checked or considered as follows:
1. line 59: “and”? check this line.
2. Line 71: “Or the purpose of restoring regional landscape connectivity.”? check this line.
3. Line 167:Is there a basis for the classification of ” 5 different scales of species dispersal scale” ?
4. Line 188-188: “forest land, shrubs and grasslands were extracted as the foreground of MSPA,” why water or wetland are not include,why birds are not be considered in 5 ecological group?
5. Line 236-238: Actually, a corridor can link two sources, how to understand “the corridor that crosses the source will cut the ecological community”?
6. Line 314. “2.2.4.1 Multiscale ecological group identification”, why the “The optimal modularity method” can be used to identify the ecological group, what is the index means to ecological group? It should be explained more.
7. Line 355: where is the “formula 4.3”?
8. Line 438: check this line.
9. Line 453:check this line.
10. Line 479:what is the ecological meaning of ” the closed loops in the ecological network gradually increase.”?
11. Line494: check this line.
12. Line 552-553: check these lines.
13. Line 603: check this line.
14. According to the 2.2.1.1. “Instead of using a single species to represent a specific group, the study used whole numbers to represent their general relationship”, the dispersal distance of species is not the actual distribution space of a species, but a number standard to classify the ecological networks, so the ecological networks identified are not equivalent to the species dispersion space. Therefore, the analysis of “the species dispersion scale”, “ ecological groups”, and ” ecological network” should be more clarified. For example, line 116 “at different species dispersion scales” or “at different spatial scales”? line 526, “ecological network at the dispersion scale of the five species” or ”ecological network at five scales” ?
15. The description of the results is too much, it is better to simplify it.
16. Line 698: The discussion of “4.3. Analysis of ecological group evolution events at multiple scales” are more like result description, lacking the further discussion about the cause or influence.
17. The research conclusions should respond to the research objectives mentioned at the Introduction.
18. What is the spatial relationship between Ecological Group boundary and ecological corridor? Is there crossover or overlap? Whether it has ecological and practical significance.
Round 2
Reviewer 5 Report (New Reviewer)
I recommend acceptance as the authors have addressed my concerns.
This manuscript is a resubmission of an earlier submission. The following is a list of the peer review reports and author responses from that submission.
Round 1
Reviewer 1 Report
Dear Editor and Authors,
I carefully read the paper "Research on Multi-Scale Ecological Network Connectivity Taking the Guangdong-Hong Kong-Macao Greater Bay Area as An Example". Bellow some general and more specific remarks.
General remarks
The structure of the text is not well organized and the writing greatly improved (e.i. Aconim explenation, citations of references, pending sentences). This does not favour readebility of manuscript.
The introduction offers a very limited background to the topics covered. In particular, the aspect of the species pools identified in the ecological network should be improved and addressed to the objectives of the work. The analysis of the topic is supported by a few number of references, in particular the reported reference18 makes an in-depth review considering the ecological point of view and reported many concepts used in the text. The authors should clarify if in their study case the approach to the ecological network analysis is based on structural or functional aspects. From my point of view the focus of the research is to quantitatively analyze the importance of nodes of the network and try to measure the general effect of them inside the network. In the manuscript I found that the functional dimension is considered only in relation to a few input parameters on species. So what is the role of this ecological group identification from the spatial point of view ? The concept should be further explored also in relation with the community concept. Deepening these aspects will allow Authors to better define the aim of the research. I suggest some papers, but there are many other very recent researches in literature to check:
Impacts of land management practice strategy on regional ecosystems:Enlightenment from ecological redline adjustment in Jiangsu, China, https://doi.org/10.1016/j.landusepol.2022.106137
Understanding ecological groups under landscape fragmentation based on network theory https://www.sciencedirect.com/science/article/abs/pii/S0169204621000293?via%3Dihub
Landscape configuration and composition shape mutualistic and antagonistic interactions among plants, bats, and ectoparasites in human-dominated tropical rainforests https://www.sciencedirect.com/science/article/pii/S1146609X21000680?casa_token=g9U-EZDavqoAAAAA:j455MB-NeUyfYmAxgdB5tS2T8r7_JKHl9unQmAl698L-5gLjqTpK0RaJionapd96qc810h79pw
In my opinion one of the main problem of the manuscript is that research aim is not clear. In the introduction Authors reports “this study takes the Guangdong-Hong 94 Kong-Macao Greater Bay Area as an example to construct an ecological network of the 95 dispersion scale of five species from 1990 to 2020 based on the Morphological Spatial Pat-96 tern Analysis (MSPA) method”, but then, the Technology Roadmap (e.g. the methodological approach graph) showed a complex approach including ecological network and ecological group identification, connectivity change analysis and ecological group evolution analysis. It would be very useful to highlight which are the real innovative aspects that the research discover. In fact, many results are reported, also taken up again in the conclusions, but the advancement in terms of new knowledge produced (expecially from the methodological point of view) by the research does not emerge.
Specific comments
MSPA was used to identify ecological sources. Authors identify this step as a weak point to be improvement in the future. Anyway a further explanation of the criterias adopted to reach the ecological network model is needed. How autoecology of the species indicated in the annex were considered to inform the MSPA system? Were the characteristics of the species (reported in the annex) considered on average or individually? So it was developed different networks for each specie or only one network for all the group of species adopted for each explored scales?
Morphological Spatial Pattern Analysis (MSPA) implied the definition of different weights which were assigned to the calculated resistance factors (Table 2). What was the criteria used for they definition and what influence they may have on the model developed?
Sentences in lines 57-59 are not clear. What does authors mean with biological habitat network system? The habitat includes the biological component (animals and plants) and the physical component (soil, etc.).
Line 132-133 According to the research data of Harestad and Bunnell, Hoffman, Grassman, Rajaratnam, etc., 13 terrestrial mammal categories with clear habitat range were further identified.Are the references missing?
Line 137-138.” Foreign scholars have divided nature reserves into large, medium and small scales according to the maximum dispersal distance of species to analyze the connectivity differences of their network of nature reserves [28]” Why foreing scolars? Further more, the reference 28 doesn't consider nature reserves but presents a methodological approach very similar to the one applied by Authors in this research. It also contains many concepts taken up in the research. The citation of this research should be enhanced.
Reviewer 2 Report
The manuscript is based on the assessment of Multi-Scale Ecological Network Connectivity for Guangdong-Hong Kong-Macao Greater Bay Area, which reflects the changing status of the Guangdong-Hong Kong-Macao Greater Bay Area.
This is a well-researched attempt but, unfortunately, the authors have not paid much attention to discussing the results correctly. A few questions arise during the review of the MS as follows:
1. How your study extends our current understanding of the phenomenon under investigation?
2. How do findings complement existing empirical insights?
3. How can estimate the changing rate in the study area?
4. What could be suggestive outcomes?
5. What are the management aspect and factors responsible for the changes or something else?
However, the methodology descriptions are clear and well-explained. Still, the language editing needs improvement throughout the MS. Few sentences are repeated in the different sections, which indicates that the authors did not attempt to write it properly.
Please find attached a couple of observations, along with recommendations
for improvement of this manuscript.

Reviewer 3 Report
Dear Authors
your work seems to be very interesting to me. However, going though it, I found that it might be enhanced using the tools ( landscape metrics), which landscape ecology provides it with us.
You have deeply addressed the issue of fragmentation in the natural environment of the agglomeration, however you did not applied to calculate to what extent the study area has been fragmentated by human intervention during 1999- 2020.
Landscape ecology provide us with landscape metrics, by which researchers are enabled to calculate structure, composition and configuration of the landscapes. For this prpose, those landscape metrics , which are belongs to the landscape configuration can be applied. There.fore, I do recommend to apply these landscape metrcs to clarify to what extent your landscape has been fragmented
Moreover, in the section of methodology, to me it is not clear to you make the ecological network, nodes, connection (corridors), how the movement resistance and the probablity have been estimated. Hence, I do recommend you to revise this part of thesection of methodology, aing at making it more clear to the possible readers of your manuscript.
Moreove, I do recommend to have your manuscript editted by an English native speaker, as there are several grammatical and structural errors in the manuscript. I have highlighted some of them.
Regarding the abstract, I would write that it can not stand alone. The abstact should be a summary of the all sections of the manuscript so that a potential reader could catch the general idea and information, which has included in the manuscript. So, I do recommend you re-written this part of the manuscript.
Concerning the introduction, I am not convinced that your work can fill the possible gap, which there are in the field of landscape fragmentation, landscape connectivity, and some thins like those. Hence, I do recommend you re-written your introduction section, aiming at clarifying what gap/gaps you have explored in the science of landscape fragmentation. Moreover, at the ned of the introduction section, it should clearly mention what objective/ objectives you are looking for achieving them by conducting the present study.
At the ending point of the manuscript, I do recommend you add a new section under titled of " Plannig Implication", under which it would be great if you might address the potential applications of your findings in planning section of the relatted adminstration of the agglomaration.
Reviewer 4 Report
General comments:
This paper takes the Guangdong-Hong Kong-Macao Greater Bay Area as an example, constructs an ecological network of the dispersion scale of five species from 1990 to 2020 based on the MSPA method, identifies the ecological groups in the network, and uses the core node-based community evolution path tracking algorithm to analyze the ecological groups. To explore the changes of ecological network connectivity at different scales in the region, and to reveal the overall and local characteristics and changes of the migratory space of terrestrial mammals with different dispersion capabilities. The paper topic and objectives are good and interesting. This paper is of great significance and value in the fields of ecological network, ecology, landscape ecology, and urban planning. However, there are still some minor problems in this article that need to be revised or further discussed.
Specific Comments:
Line 18: When MSPA first appeared in the manuscript, it was suggested to use the full name, which may not be known to scholars in other fields.
Line 120: Where is Table 1?
Line 149: Where is Supplementary Materials Table 1?
Line 150: (2) should be changed to 2.2.1.2.?
Line 172 & 471: There are two “Tables 2”.
Line 185: Figure 3. has the same name as Figure 2.
Line 399 & 402: Where are Figures in Supplementary Material?
Line 629: The legend in Figure 10 is ambiguous.
References: There are few references on ecological network or landscape ecological network, especially in the last three years.
